# Alcids 'fly' at efficient Strouhal numbers in both air and water but vary stroke velocity and angle

Anthony B Lapsansky[1]*, Daniel Zatz[2], Bret W Tobalske[1]

[1]Field Research Station at Fort Missoula, Division of Biological Sciences, University of Montana, Missoula, United States; [2]ZatzWorks Inc, Homer, United States

**Abstract** Birds that use their wings for 'flight' in both air and water are expected to fly poorly in each fluid relative to single-fluid specialists; that is, these jacks-of-all-trades should be the masters of none. Alcids exhibit exceptional dive performance while retaining aerial flight. We hypothesized that alcids maintain efficient Strouhal numbers and stroke velocities across air and water, allowing them to mitigate the costs of their 'fluid generalism'. We show that alcids cruise at Strouhal numbers between 0.10 and 0.40 – on par with single-fluid specialists – in both air and water but flap their wings ∼ 50% slower in water. Thus, these species either contract their muscles at inefficient velocities or maintain a two-geared muscle system, highlighting a clear cost to using the same morphology for locomotion in two fluids. Additionally, alcids varied stroke-plane angle between air and water and chord angle during aquatic flight, expanding their performance envelope.

## Introduction

The 'jack of all trades' concept – the idea that the ability to function in multiple environments can only be achieved by sacrificing maximal performance (*MacArthur, 1972*) – is commonly invoked in discussing the locomotor performance of wing-propelled diving birds (*Elliott et al., 2013*; *Simpson, 1946*; *Stettenheim, 1959*; *Storer, 1960*; *Thaxter et al., 2010*). These species, which include some or all members of the alcids (*Alcidae*), ducks (*Anatidae*), petrels and shearwaters (*Procellariidae*), dippers (*Cinclus*), and the penguins (*Spheniscidae*), use their wings to propel themselves underwater. Wing-propelled diving birds which have retained their ability to fly in the air – hereafter, 'dual-medium' species (sensu *Kovacs and Meyers, 2000*) – are fluid generalists. These animals use the same locomotor apparatus to 'fly' in both air and water, and are, therefore, expected to fly poorly relative to strictly aerial and strictly aquatic fliers in each environment.

Interestingly, birds in the family Alcidae (puffins, murres, and their relatives) seem to contradict the notion of a trade-off between aerial and aquatic flight performance. As with many dual-medium birds, alcids have high wing-loading (the ratio of body mass to wing area), and therefore display poor maneuverability in aerial flight relative to non-diving birds (*Ortega-Jimenez et al., 2011*; *Shepard et al., 2019*). However, the wing-loadings of alcids and other dual-medium birds are nearly indistinguishable from those of volant birds which use their feet for aquatic locomotion (based on data from *Alerstam et al., 2007*; *Bruderer et al., 2010*; *Spear and Ainley, 1997*), indicating that high wing-loading is likely the result of selection by the aquatic environment for large body sizes or low buoyancy (*Ponganis, 2015*), rather than a trade-off specific to dual-medium flight. The current records for the depth and duration of a single dive by an alcid are 210 m and 224 s, respectively, held by the ∼1 kg thick-billed murre (*Uria lomvia*, Linnaeus 1758), making this alcid, on a mass-specific basis, the deepest and longest-duration diver on earth (*Croll et al., 1992*). When corrected for

*For correspondence:
anthony.lapsansky@umontana.edu

body size, alcids exhibit dive durations and depths far greater than even penguins (*Halsey et al., 2006*).

One possible explanation for the aquatic performance of alcids is that they have mitigated the costs of dual-medium flight. Specifically, by maintaining efficient Strouhal numbers (*St*) and stroke velocities across air and water, birds in the family Alcidae may lessen the perceivable differences between aerial and aquatic flight, thereby reducing the costs of fluid generalism.

To swim or fly, an animal must impart momentum to the surrounding fluid. Strouhal number ($St = fAU^{-1}$, where *f* is wingbeat frequency, *A* is wing excursion, and *U* is forward speed) describes the pattern of vortices shed into the fluid wake by a flapping foil as it imparts that momentum (*Triantafyllou et al., 1993*). Extensive research has determined that peak efficiency (in terms of the power required to flap a foil relative to the thrust output to the fluid) for a simple heaving and pitching foil occurs at around $0.2 < St < 0.4$ (*Anderson et al., 1998*; *Triantafyllou et al., 1991*; *Triantafyllou et al., 1993*). Most flapping and swimming animals studied to date fall within or near that range, with the previously studied, strictly aerial birds exhibiting $0.12 < St < 0.47$ during cruising flight (*Taylor et al., 2003*). That most species fall near the efficient range of *St* suggests that natural selection has tuned the kinematics of animals to fly and swim efficiently (*Nudds et al., 2004*; *Taylor et al., 2003*). Thus, alcids could achieve efficient fluid wake production in both air and water by maintaining $0.2 < St < 0.4$, and efficiency on par with previously studied single-media species by maintaining $0.12 < St < 0.47$, but the 'jack of all trades' concept suggests that they may unable to do so given the substantial differences in density and viscosity between the two fluids.

Stroke velocity describes the speed at which the wing is swept through its arc during either the downstroke or the upstroke of the wing. This parameter is likely important in determining the cost of locomotion given that it should be proportional to the contractile velocity of the major flight muscles, the pectoralis and the supracoracoideus (*Hamilton, 2006*; *Tobalske and Dial, 1994*; *Tobalske et al., 1999*). Muscles fibers of a given fiber type and myosin isoform are most efficient at converting metabolic power into mechanical power over a narrow range of contractile velocities (*Goldspink, 1977*; *He et al., 2000*; *Reggiani et al., 1997*; *Rome et al., 1988*). Thus, it would behoove alcids to operate the fibers in their flight muscles at the contractile velocity which maximizes muscle efficiency, and for that velocity to be shared across aerial and aquatic flight. Otherwise, alcids could maintain two populations of fibers – an aerial set and an aquatic set – but this would add mass to the animal, increasing the cost of aerial flight (*Ellington, 1984a*). Previous research has demonstrated that diving alcids maintain stroke velocities within a narrow range across dive depths, despite large variations in buoyancy, suggesting that they are responsive to the challenge of maintaining contractile velocity (*Watanuki and Sato, 2008*; *Watanuki et al., 2006*). Although researchers have not yet examined myosin isoforms in alcids, the two species of alcids for which histochemical data are available possess only 'fast' muscle fibers (*Kovacs and Meyers, 2000*; *Meyers et al., 1992*).

Recently, *Kikuchi et al., 2015* measured the kinematics of flying and diving rhinoceros auklets (*Cerorhinca monocerata*) using a combination of videography and accelerometry. The authors used bootstrapping to coalesce measurements from various individuals to determine the range of *St* exhibited by this species. The results of this study strongly suggest that this small alcid maintains optimal *St* in air and water. We wanted to extend this work to determine if individuals tune their kinematics to match optimal *St* on a per-flight basis. Alternatively, it is possible that the average kinematics of this species are simply centered between $0.2 < St < 0.4$. These authors also suggest, based on wingbeat frequency, that stroke velocities of rhinoceros auklets are lower in water, but were unable to statistically compare stroke velocities during aerial versus aquatic flight. While wingbeat frequencies are different between aquatic and aerial flight (~2–4 Hz versus 7–11 Hz, respectively), wingbeat amplitude may vary between the two environments, allowing for similar stroke velocities.

To improve understanding of potential evolutionary trade-offs between aerial and aquatic flight, we tested whether alcids exhibit efficient *St* and maintain consistent stroke velocities when flying in water and air. We used videography to measure the wing kinematics of four species of alcids from three genera. These species differ substantially in body mass (450 g to 1 kg) and represent opposite branches of the alcid phylogeny. In addition to *St* and stroke velocity, we report a variety of kinematic parameters, including stroke-plane and chord angles relative to the body to contrast how the flight apparatus is used in air versus water and during horizontal versus descending aquatic flight.

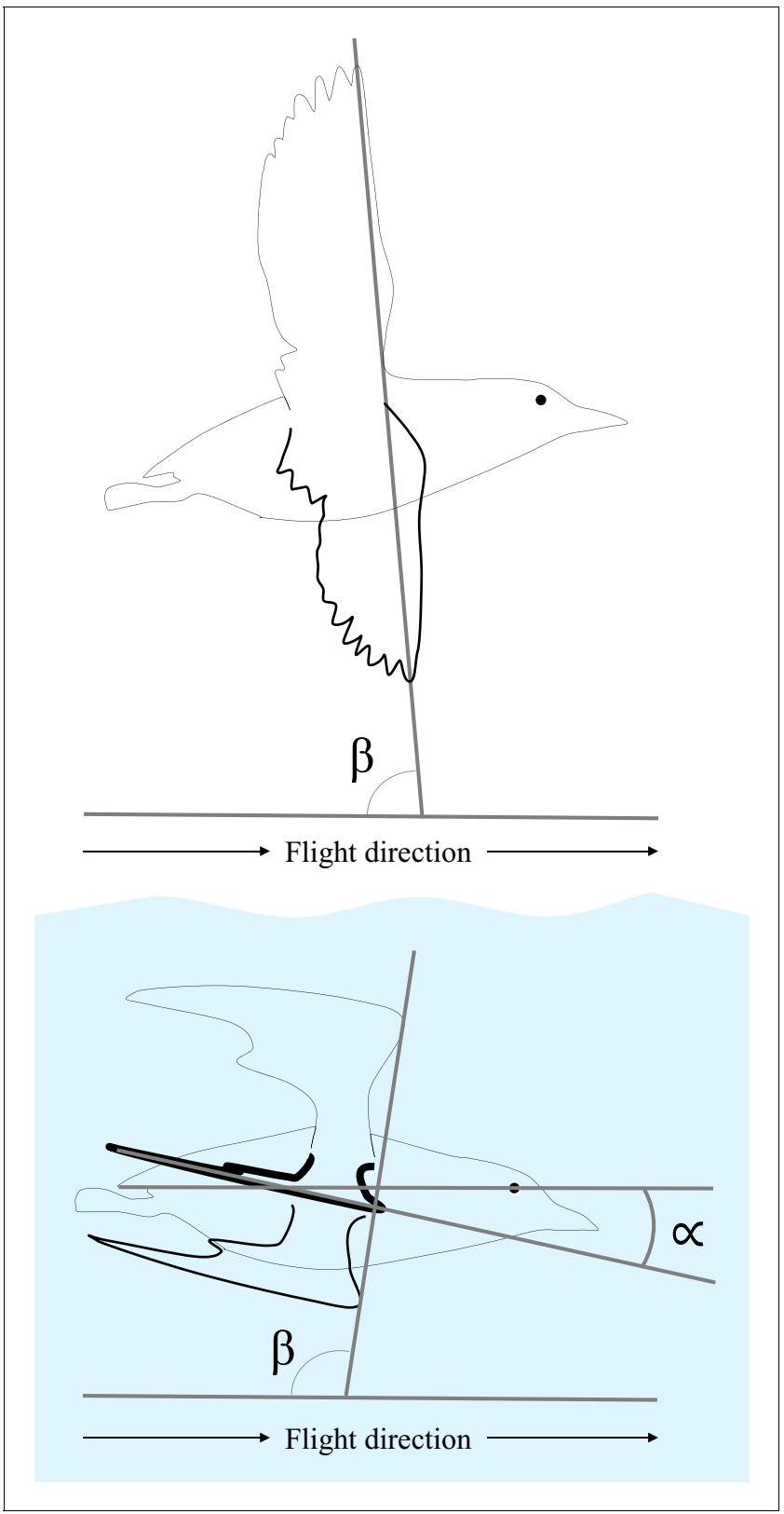

**Figure 1.** Measurements of stroke-plane angle (β) and chord angle (∝). The wings drawn with the thin black line indicate the position at the start of downstroke in air (top) and water (bottom, with blue shading). The wings drawn with the dashed line indicate the position at the end of downstroke in air and water. β was measured using the wingtip in aerial flight and the wrist in aquatic flight. ∝ was measured at mid-upstroke and mid-downstroke (wing drawn with thick black line) during aquatic flight.

We hypothesized that alcids maintain efficient $St$ and consistent stroke velocities across air and water, which would allow this group to mitigate the costs of fluid generalism (*Figure 1*).

## Results

Strouhal numbers ($St$) for horizontal aquatic flights averaged 0.18 ± 0.02 for common murres, 0.13 ± 0.01 for horned puffins, 0.15 ± 0.01 for pigeon guillemots, and 0.15 ± 0.02 for tufted puffins (*Figure 2*, blue points). $St$ for descending aquatic flights were significantly greater than those for horizontal aquatic flights ($F_{1,27}$ = 145.6, $\eta^2$ = 0.729, p-value = 2.18e-12) with a relatively minor but significant interaction between species and the type of aquatic flight ($F_{3,27}$ = 3.59, $\eta^2$ = 0.054, p-value = 0.0264). Within-species *post hoc* tests indicated that all species exhibited greater $St$ during descending aquatic flights relative to horizontal aquatic flights (p = 0.0478, 4.19e-04, 9.17e-07, 5.72e-07; for species in alphabetical order). $St$ for descending aquatic flights averaged 0.24 ± 0.01 for flights of common murres, 0.21 ± 0.04 for horned puffins, 0.29 ± 0.06 for pigeon guillemots, and 0.29 ± 0.03 for tufted puffins (*Figure 2*, green points).

$St$ for aerial flights based on the ground speed of the birds averaged 0.17 ± 0.02 for common murres, 0.22 ± 0.14 for horned puffins, 0.49 ± 0.06 for pigeon guillemots, and 0.27 ± 0.02 for tufted puffins (*Figure 2*, dark red points). Except for the flights of common murres, all birds appeared to be flying in considerable wind based on the size of the waves on the surface of the water. Thus, we also calculated $St$ for aerial flights based on the airspeed characteristic of each species as reported in *Spear and Ainley, 1997* (see Materials and methods for details). When estimated from the range of measured cruising flight speeds, $St$ for aerial flights ranged from 0.12 to 0.25 for flights of common murres, 0.13 to 0.27 for horned puffins, 0.18 to 0.27 for pigeon guillemots, and 0.16 to 0.25 for tufted puffins (*Figure 2*, light red lines).

Downstroke velocities were significantly greater during aerial flights than during aquatic flights for all four species (*t-Value* = 8.10, 11.5, 6.04, 25.9; *df* = 16.5, 19.0, 9.48, 16.5; p-values = 3.80e-07,

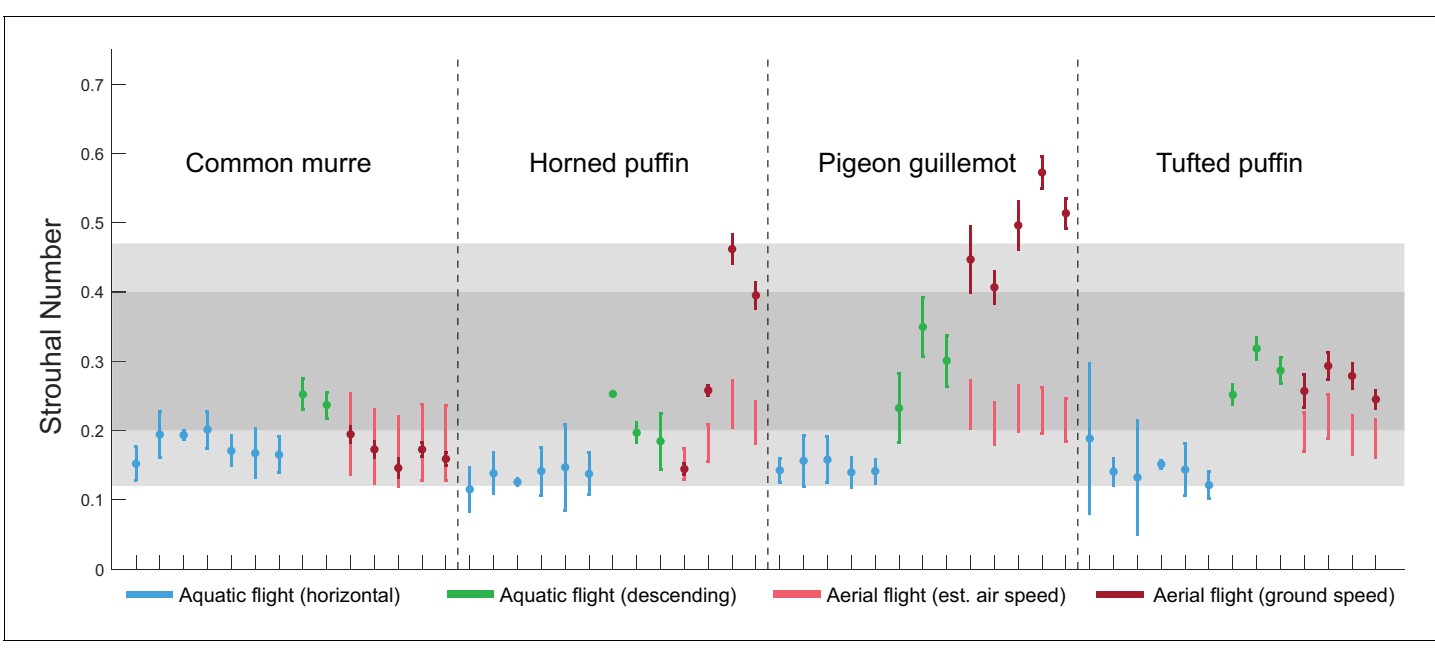

**Figure 2.** Strouhal numbers ($St$) of four species of alcid in aerial and aquatic flight. Each hatch mark on the x-axis indicates a unique flight. The darker shaded section indicates 0.2 < $St$ < 0.4, in which propulsive efficiency is predicted to peak, and the lighter shaded region indicates 0.12 < $St$ < 0.47, which is the range of St exhibited during cruising flight of strictly aerial birds reported in *Taylor et al., 2003*. Points indicate $St$ for horizontal aquatic flights (blue), descending aquatic flights (green), aerial flights based on ground speed (dark red), and aerial flights calculated using the range of cruising speeds of that species reported in the literature (light red). Each flight is represented by the mean $St$ for that flight ± s.d., except for $St$ calculated for aerial flights based on airspeed, for which we chose not to indicate a central tendency.

The online version of this article includes the following source data for figure 2:

**Source data 1.** Strouhal numbers of four species of alcid in aerial and aquatic flight.

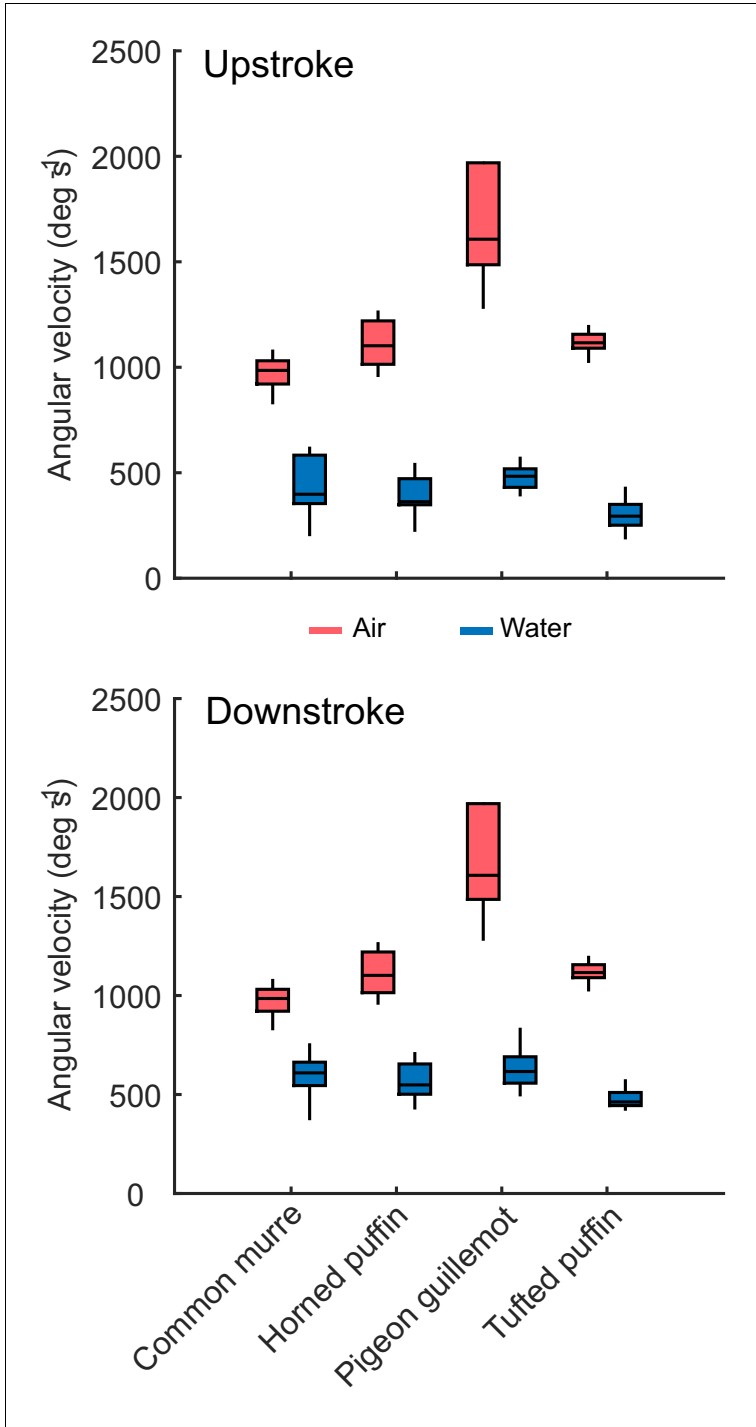

**Figure 3.** Stroke velocities of four species of alcid in aerial and aquatic flight. Stroke velocity was significantly greater during aerial flights (red) than during aquatic flights (blue) for each of the four species for both downstroke (t-Value = 8.10, 11.5, 6.04, 25.9; df = 16.5, 19.0, 9.48, 16.5; p-values = 3.80e-07, 5.19e-10, 1.55e-04, 8.56e-15; for species in alphabetical order) and upstroke (t-Value = 10.5, 16.0, 6.83, 26.4; df = 15.0, 18.8, 9.18, 16.1; p-values = 2.67e-08, 2.13e-12, 6.97e-05, 1.09e-14; for species in alphabetical order). The central line in each box marks the median, while the upper and lower margins of the box indicate the quartile range. The entire range of values lie between the whiskers.

The online version of this article includes the following source data for figure 3:

**Source data 1.** Stroke velocities of four species of alcid in aerial and aquatic flight.

5.19e-10, 1.55e-04, 8.56e-15; for species in alphabetical order; *Figure 3*). The same was true for upstroke velocities (*t-Value* = 10.5, 16.0, 6.83, 26.4; *df* = 15.0, 18.8, 9.18, 16.1; p-values = 2.67e-08, 2.13e-12, 6.97e-05, 1.09e-14; for species in alphabetical order; *Figure 3*). Wingbeat amplitudes were greater during aquatic flights across species ($F_{1,71}$ = 162.4, $\eta^2$ = 0.597, p-value<2.2e-16; *Figure 4*) with a relatively minor but significant interaction between species and fluid ($F_{3,71}$ = 3.52, $\eta^2$ = 0.039, p-value=0.019). Within-species *post hoc* tests indicated that all four species exhibited significantly greater wingbeat amplitudes during aquatic flight relative to aerial flight (p-values=1.93e-06,<1.0e-08, 3.29e-04, 2.36e-05). Stroke durations were often 2-3X greater in water as compared to air, as indicated by the differences in wingbeat frequency (*Figure 4*), leading to significant differences in stroke velocities between fluids.

When horizontal and descending aquatic flights are grouped together, stroke-plane angle (β) was significantly lower (the top of stroke plane is rotated more caudally) during aerial flights relative to aquatic flights ($F_{1,47}$ = 41.3, $\eta^2$ = 0.422, p = 6.14e-08; *Figure 5*). Across species, stroke-plane angle averaged 79 ± 7 deg for aerial flights, 92 ± 7 deg for horizontal aquatic flights, and 93 ± 12 deg for descending aquatic flights. Within aquatic flights, there was no significant relationship between stroke-plane angle and angle of descent ($F_{1,27}$ = 0.0755, $\eta^2$ = 0.002, p = 0.786; *Figure 5*).

There was a significant relationship between chord angle (α) and angle of descent for upstroke ($F_{1,30}$ = 55.7, $\eta^2$ = 0.458, p = 2.55e-08; *Figure 6*) and downstroke ($F_{1,27}$ = 8.17, $\eta^2$ = 0.122, p = 8.11e-03; *Figure 6*). However, a significant crossed interaction between species and angle of descent for downstroke chord angle ($F_{3,27}$ = 7.68, $\eta^2$ = 0.343, p = 7.26e-4), indicates that the main effect of angle of descent on chord angle during downstroke is uninterpretable (i.e. the response depends on the species; *Figure 6*). However, alcids significantly increased chord angle (thus, the degree of supination) during upstroke as a function of angle of descent.

## Discussion

Alcids achieve efficient wake production based on *St* during both aerial flight (based on airspeed) and during aquatic flight. While *St* for horizontal aquatic flights often fell below *St* = 0.2 (*Figure 2*, blue points), all measured values overlapped with the range for the cruising aerial flight of strictly aerial birds reported in the literature (*Taylor et al., 2003*). Because stroke velocities were substantially different between air and water (*Figure 3*), the use of efficient *St* seems to come at a cost to the contractile efficiency of the primary flight muscles. Alternatively, aerial and aquatic flight may be powered by different sets of muscles, as discussed below.

We interpret the relatively low values of *St* during horizontal aquatic flight to be a consequence of buoyancy. While swimming horizontally, alcids must counteract buoyancy as it attempts to pull them toward their dorsal side. Buoyancy is especially strong at shallow depths, as air volumes compress with depth (*Wilson et al., 1992*). To compensate for buoyancy during horizontal aquatic flight, alcids in this study seemed to produce quick, low excursion wingbeats with near-horizontal chord angles (α) on the upstroke (*Figure 6*). Given that the upstroke produces negative heave (ventrally directed acceleration) in swimming alcids (*Watanuki et al., 2006*), these kinematics seem to be a strategy used to counteract the strong, dorsally oriented buoyancy experienced during horizontal swimming at shallow depths. In contrast, descending alcids must counteract buoyancy as it attempts to resist their forward motion and are, therefore, not required to produce negative heave via the upstroke. Still, all values of *St* for horizontal aquatic flight overlapped with the range reported for strictly aerial birds in aerial flight – 0.12 < *St* < 0.47 – (Figure 1), suggesting that alcids produce wakes of similar efficiency to their fully aerial relatives even while fighting buoyancy.

From previous research, the precise range of *St* values which confer optimal propulsive efficiency (the proportion of total mechanical energy expended that contributes to useful work) depends somewhat on the kinematics of the flapping foil, but departures from that range can have substantial effects (*Anderson et al., 1998*; *Read et al., 2003*). Data comparing *St* to propulsive efficiency in animals are limited, but *Rohr and Fish, 2004* report that a relatively minor shift in *St* in cetaceans (e.g. from 0.25 to 0.35 in *Pseudorca crassidens*) can reduce propulsive efficiency by 5–10% (*Rohr and Fish, 2004*). The paucity of data for animals swimming and flying outside the optimal range of *St* may be due to the challenge of eliciting inefficient kinematics from animals. Alternatively, because translational velocity is partially determined by wing excursion and frequency, the convergence of *St* on some range of values may be inevitable. The latter seems unlikely, however, as trout adhere to a

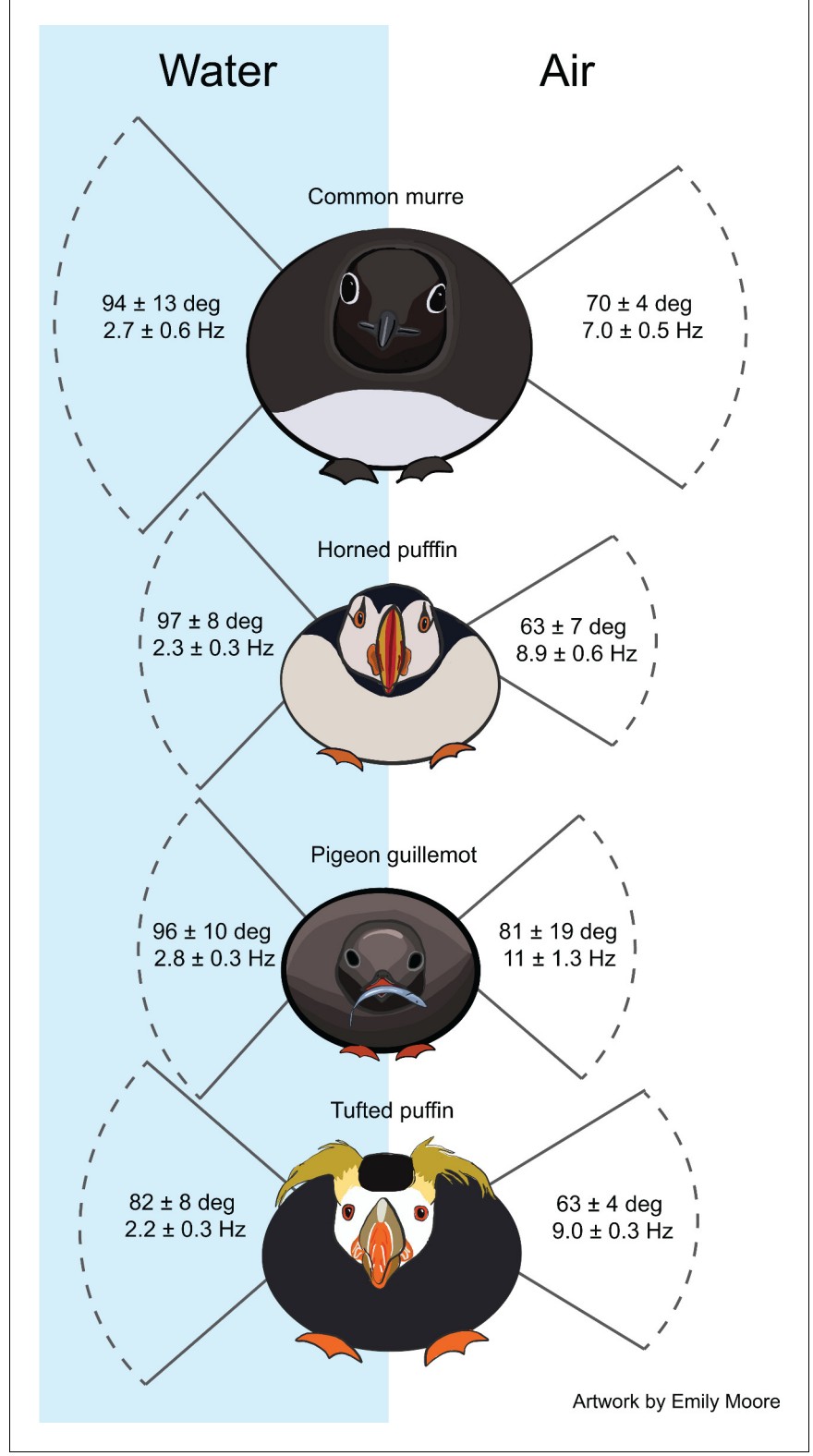

**Figure 4.** Wingbeat amplitude and frequency of four species of alcid in aerial and aquatic flight.

Artwork by Emily Moore

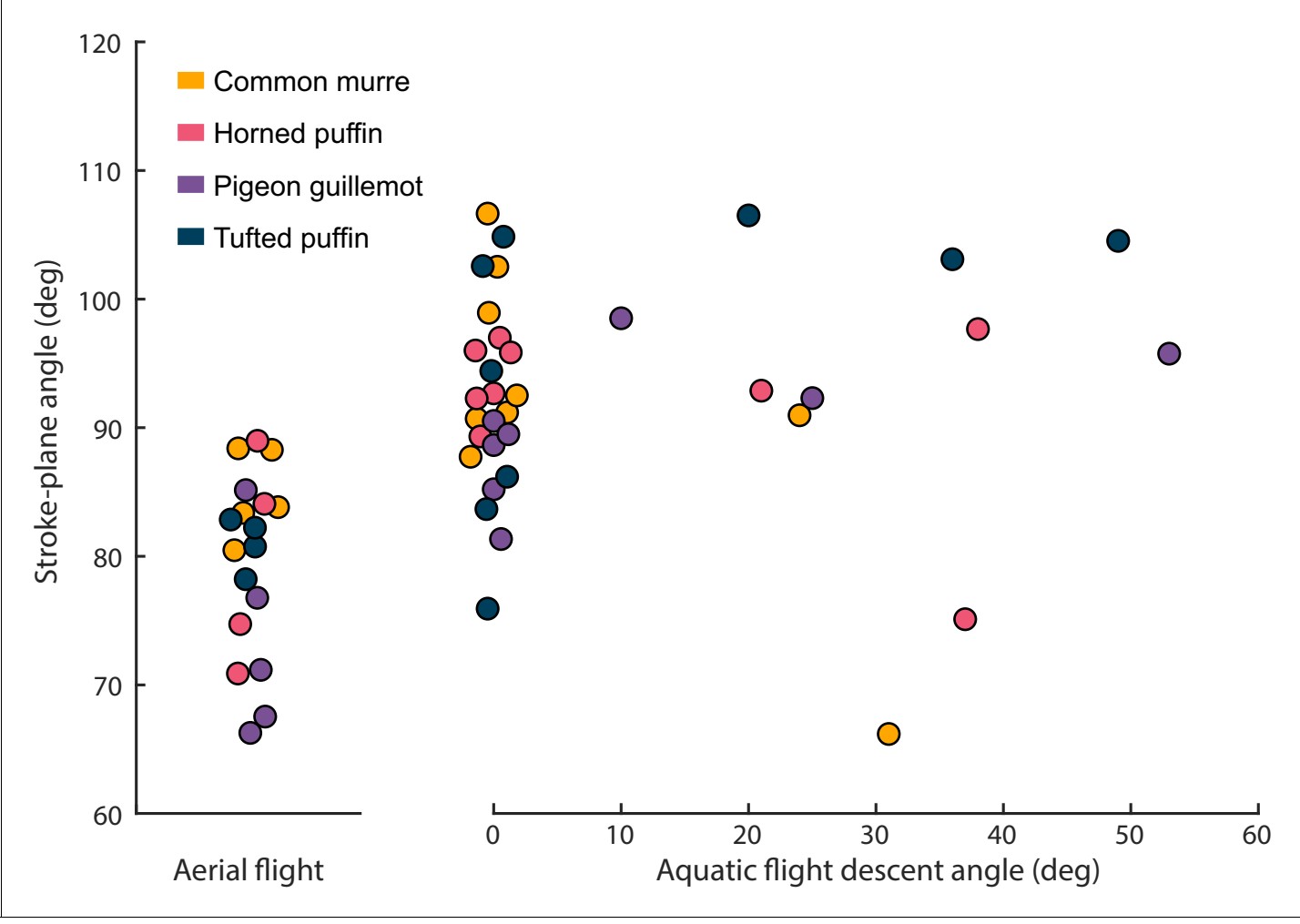

**Figure 5.** Stroke-plane angle (β) of four species of alcid in aerial and aquatic flight. β was significantly lower (the top of the stroke plane was rotated more caudally) during aerial flights relative to aquatic flights ($F_{1,47}$ = 41.3, $\eta^2$ = 0.422, p = 6.14e-08). Within aquatic flights, there was no consistent relationship between β and the angle of descent ($F_{1,27}$ = 0.0755, $\eta^2$ = 0.002, p = 0.786). Jitter was added to the points representing aerial flights and horizontal aquatic flights (descent angle = 0) to increase visibility.

The online version of this article includes the following source data for figure 5:

**Source data 1.** Stroke-plane angle of four species of alcid in aerial and aquatic flight.

narrow range of *St* despite experimentally-induced disruptions to their kinematics (*Nudds et al., 2014*).

One limitation of this study is that recordings of aquatic flight were made at shallow depths. However, previous work has indicated that velocity (*Lovvorn et al., 2004*; *Watanuki and Sato, 2008*; *Watanuki et al., 2006*), wingbeat frequency (*Watanuki et al., 2006*), and wing excursion (*Kikuchi et al., 2015*) of descending birds remain within a narrow range across depths, suggesting that our results apply to wild alcids. Average velocity and wingbeat frequency of common murres during swimming in this study were 1.63 m s$^{-1}$ and 2.4 Hz, respectively, whereas *Watanuki et al., 2006* report 1.61 m s$^{-1}$ and 2.6 Hz for wild birds (*Watanuki et al., 2006*).

In moving between air and water, alcids must cope with a dramatic shift to the forces exerted upon them. For example, a bird in aerial flight must counteract the downward pull of gravity, whereas the same bird in shallow water must counteract the upward pull of buoyancy. Recent work with robotics has revealed that a simple shift in stroke-plane angle (β, *Figure 1*) can allow for both aerial and aquatic propulsion from the same wing (*Izraelevitz et al., 2018*). The authors of this study point to alcids as their inspiration for exploring stroke-plane angle in a hybrid, flapping wing, but, to

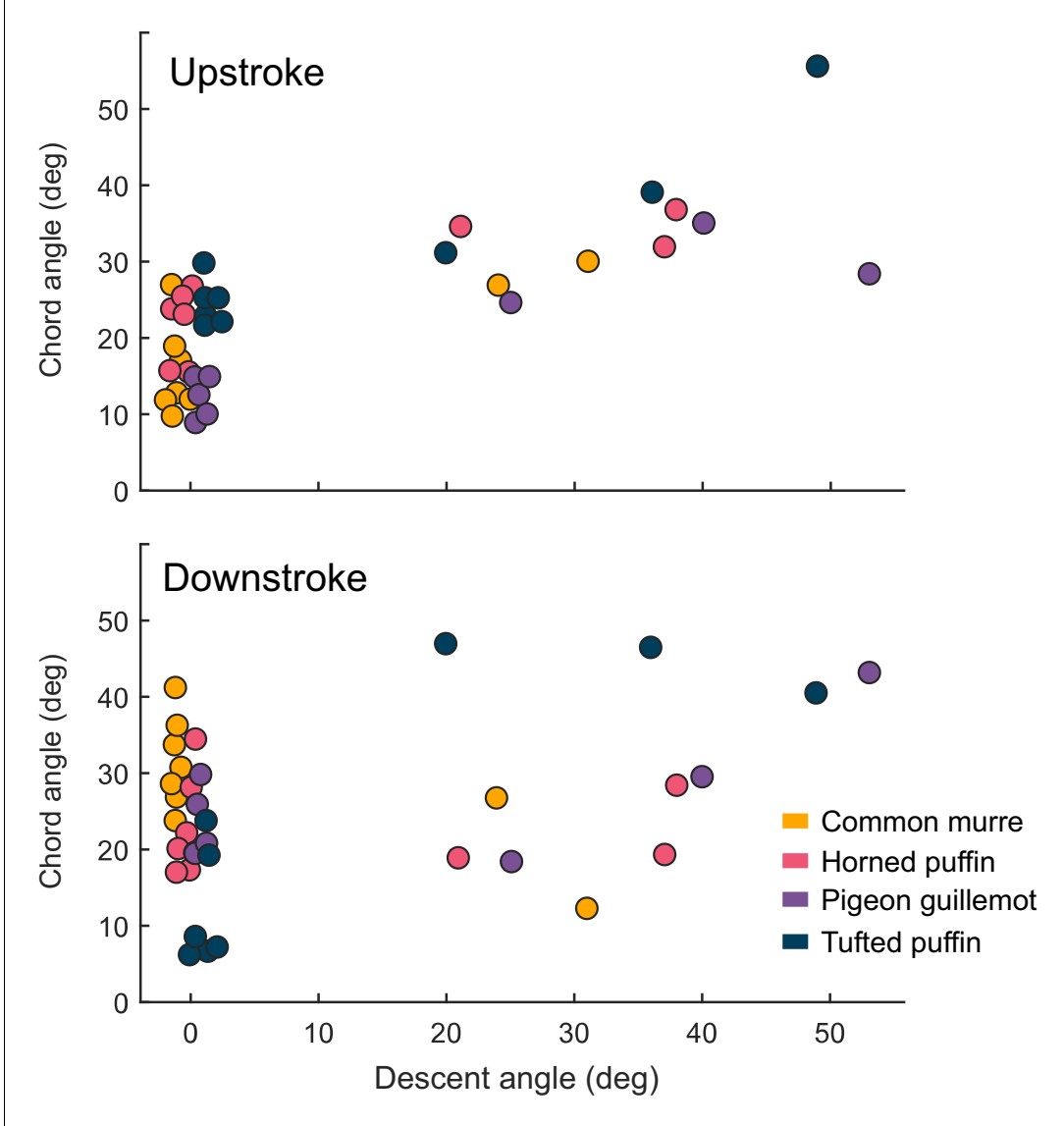

**Figure 6.** Chord angle (α) versus descent angle for aquatic flights of four species of alcids. α increased with the angle of descent for upstroke ($F_{1,30}$ = 55.7, $\eta^2$ = 0.458, p = 2.55e-08) and downstroke ($F_{1,27}$ = 8.17, $\eta^2$ = 0.122, p = 8.11e-03). However, a significant crossed interaction between species and angle of descent for downstroke chord angle ($F_{3,27}$ = 7.68, $\eta^2$ = 0.343, p = 7.26e-4), indicates that the main effect of angle of descent on chord angle during downstroke is uninterpretable (i.e. the response depends on the species). Jitter was added to the points representing horizontal aquatic flights (descent angle = 0) to make all points visible.

The online version of this article includes the following source data for figure 6:

**Source data 1.** Chord angle versus descent angle for aquatic flights of four species of alcids.

our knowledge, our result is the first to confirm that β varies as predicted by *Izraelevitz et al., 2018* as dual-medium birds transition between air and water.

The stroke plane is rotated to a greater degree during aerial flight to values that are consistent with strictly aerial fliers (*Figure 5*; *Tobalske et al., 1999*). In other words, during the aerial down-stroke, while the wing is being depressed alcids also draw the wing forward. To reset the position, alcids elevate and retract the wing during the aerial upstroke. According to *Izraelevitz et al., 2018*, this stroke-plane angle helps create the vertical force needed to counteract gravity in air. In water, wherein a bird is actually pulled up by buoyancy rather than down by gravity, the top of the stroke plane rotates cranially (*Figure 5*), allowing the bird to orient net force production to counteract drag

(*Izraelevitz et al., 2018*). Thus, alcids shift stroke-plane angle to cope with the shift in external forces between air and water.

We found no significant relationship between the angle of descent and stroke-plane angle, suggesting that – while stroke-plane angle varies between air and water – alcids do not seem to further modify β to fine-tune the direction of their force output (*Figure 5*). Instead, alcids appear to change the orientation of their force output during aquatic flight, at least in part, by increasing upstroke chord angle (α) with angle of descent (*Figure 6*).

While amplitude was greater in water for all species (*Figure 4*), as expected, stroke durations were dramatically shorter, causing stroke velocities in aerial flight to be ~2X faster than those during aquatic flight (*Figure 3*). The work of *Kikuchi et al., 2015* suggest a similar result for rhinoceros auklets; however, they report nearly equal wingbeat amplitudes across fluids (87 deg in water, 88 deg in air). Our results may differ because we measured wingbeat amplitude in different ways. *Kikuchi et al., 2015* relied on the vertical extent of the wingtip in aerial flight and the estimated half-wingspan. Based on our observations, the excursion of the wingtip may not be a reliable measure for inferring contractile velocity. This is because the distal feathers bend considerably during the end of each half-stroke in air, increasing the perceived wingbeat amplitude. This means that stroke velocity measured via the wingtip in aerial flight is not comparable to that measured at the wrist during aquatic flight.

Assuming stroke velocity is proportional to contractile velocity of the major wing muscles, the pectoralis and supracoracoideus, alcids either contract these muscles at inefficient velocities in one or both fluids or maintain a two-geared system – with one set of muscle fibers used for aquatic flight and another for aerial flight. This is because muscle fibers of a given fiber type and myosin isoform are most efficient over a narrow range of contractile velocities (*Goldspink, 1977*; *He et al., 2000*; *Reggiani et al., 1997*; *Rome et al., 1988*). Alcids have exceptionally long sterna, perhaps allowing for regional specializations in the pectoralis and supracoracoideus (*Hamilton, 2006*; *Kovacs and Meyers, 2000*; *Stettenheim, 1959*). Alternatively, Kovacs and Meyers indicate that the latissimus dorsi caudalis, which is enlarged in alcids, is positioned to retract the wing as occurs during the aquatic downstroke. Thus, alcids may rely on different muscles for powering the downstroke in each fluid (*Kovacs and Meyers, 2000*). Additionally, previous histology research has documented two, 'fast' fiber-types in the muscles (both with fast myosin but differing slightly in oxidative and glycolytic capacities) of Atlantic puffins (*Kovacs and Meyers, 2000*). These lines of evidence suggest the presence of a two-geared flight system, the number of myosin isoforms in these muscles in alcids or their contractile properties remain unknown.

By maintaining a two-geared system, alcids would avoid the costs of inefficient muscle contractions but would have increased aerial flight costs due to the additional mass of the 'aquatic gear' (*Ellington, 1984b*). In contrast, maintaining the 'aerial gear' may actually benefit aquatic performance, as muscle represents a vital oxygen storage site to diving animals (*Ponganis, 2015*). Consistent with this hypothesis, the metabolic rate of common murres is high in aerial flight but low in aquatic flight (*Elliott et al., 2013*). Future research should test whether the pectoralis and supracoracoideus muscles contract at different speeds in aerial and aquatic flight and explore in more detail the variation in myosin composition of those muscles to test for a two-geared system. It would be especially interesting to explore the presence of a two-geared system in dippers (genus *Cinclus*) – the only dual-medium passerine birds – given that passerines often express only one myosin isoform (*Rosser et al., 1996*).

## Conclusion

Alcids cruised within the efficient range of *St* in both aerial flight and aquatic flight, suggesting that selection has optimized these species for locomotion in remarkably different fluids. However, alcids flapped their wings at two discrete sets of stroke velocities according to fluid medium, indicating that they either contract their muscles at inefficient velocities in one or both fluids or maintain a two-geared muscle system, with one set of muscle fibers used in air and another in water. In addition, stroke-plane (β) and chord (α) angles appear to be important in allowing alcids to shift the orientation of their force output between media and among descent angles in water. Future research should explore the potential of a two-geared muscle system in dual-medium birds by examining myosin isoforms in alcids and other species and test for functional and regional specializations in the flight apparatus across dual-medium birds.

## Materials and methods

### Study area and animals

Study animals were common murres (*Uria aalge,* Pontoppidan 1763), pigeon guillemots (*Cepphus Columba,* Pallas 1811), horned puffins (*Fratercula corniculata,* Naumann 1821), and tufted puffins (*Fratercula corniculate*, Pallas 1769).

Filming of aquatic flight was performed at the Alaska SeaLife Center in Seward, Alaska. The Alaska SeaLife Center contains an outdoor aviary exhibit with a large area for aerial flight (approximately 20 m wide X 20 m long X 8–10 m tall) over a 397,500-liter saltwater tank. The surface of the water measures approximately 10.5 m X 11 m and is approximately 6.5 m deep at its deepest point. The southern edge of the tank is inset with a large glass viewing window approximately 3.5 m wide which extends from ~2 m above the waterline to the floor of the tank. The glass of the viewing window varies from ~6.5 cm to ~25.0 cm thick from the waterline to the floor of the tank. At the time of this study, the exhibit contained 12 horned puffins, 10 tufted puffins, 4 pigeon guillemots, and 6 common murres. Individuals of each species of alcid regularly swam past the viewing window. Birds opted either to swim parallel to the water's surface and at depths of 0.5–3 m, presumably for transportation around the tank, or to descend to the bottom of the tank. Given the clear contrasts between these two behaviors, we differentiate between horizontal (trajectory <10 deg) and descending aquatic flight (trajectory ≥20 deg). The birds swam on their own volition and selected their own swimming speeds and descent angles.

Videos of aquatic flight of all four species were taken using a GoPro Hero6 Black (GoPro, Inc, San Mateo, California, USA) at 119.88 fps and a shutter speed of 1/480 s in the 'Linear View' mode (*Video 1*, bottom panel), which removes the 'fisheye' distortion common to action cameras (Tyson Hedrick, pers. comm.). The camera was positioned on a tripod and leveled using a bubble-type level embedded in the tripod. Because birds chose when and where to dive, swimming bouts were sampled opportunistically. The camera was triggered *via* a GoPro Smart Remote (GoPro, Inc, San Mateo, California, USA) when we noticed a bird about to initiate a dive or swim past the viewing window. The camera was positioned approximately 1 m below the waterline; thus, all analyzed dives were between 0 to 3 m deep.

Videos of common murres, horned puffins, and tufted puffins in aerial flight were recorded using a Red DSMC2 with a Helium 8K S35 sensor (Red Digital Cinema, Irvine, California, USA) at 29.97 or 59.94 fps and an auto shutter (*Video 1*, top panel). The camera was attached to a Cineflex gyro-stabilizated system (General Dynamics Global Imaging Technologies, General Dynamics Corporation, West Falls Church, Virginia, United States) mounted underneath a helicopter and recorded video of birds cruising over open water in Kachemak Bay, Alaska. Videos of pigeon guillemots in *aerial flight* were recorded from land at Monterey Bay, California using a Fastec Ts5 (Fastec Imaging, San Diego, California, USA) at 239.76 fps and a shutter speed of 1/960 s. The birds flew on their own volition and selected their own speeds.

### Kinematic analyses

We performed kinematic analyses using MATLAB (2018a & b, MathWorks, Inc, Natick, Massachusetts, USA) using the DLTdv6 digitization tool described in *Hedrick, 2008* with additional analyses performed using MATLAB and IGOR Pro (v. 6.01, Wavemetrics, Inc, Beaverton, OR). Over 45,000 points were hand-digitized for this study.

### Flights perpendicular to the camera view

We gathered data on wing excursion, wingbeat frequency (Hz), bird-centered chord angle (∝) and stroke-plane angle (β), and translational velocity (body lengths s$^{-1}$) from flights of birds made perpendicular to the camera view (*Figure 1*). We were stringent in this assessment,

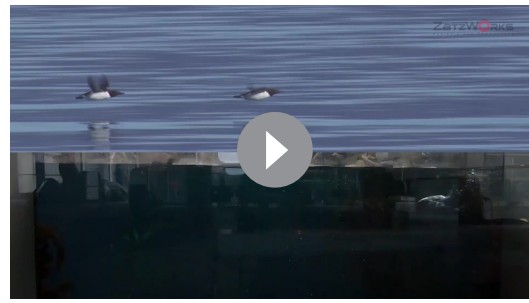

**Video 1.** Aerial and aquatic flight of the Common murre, *Uria aalge*.
https://elifesciences.org/articles/55774#video1

selecting less than 5% of all video recordings for analysis. Still, due to the nature of the cosine law, even if birds were swimming 20° off from perpendicular it would only impact our estimates of relevant kinematic parameters by about 6%. Because animals are only expected to exhibit efficient $St$ during cruising locomotion, the flight velocity of each animal was first visualized to ensure that the animal did not consistently accelerate or decelerate during a flight prior to its inclusion in our dataset.

For both aerial and aquatic flights, we digitized the eye, tip of the tail, and either the wrist (aquatic flight) or wingtip (aerial flight). The digitized points were analyzed in MATLAB using a custom script. For aquatic flights, the code first computed the angle between the bird's mean path and the waterline. If the bird was descending (trajectory >20 deg from horizontal), the code rotated, *via* a 2D Euler-angle rotation matrix, the digitized points about that angle so that the x- and y-axes were parallel and perpendicular to the bird's swimming direction, respectively. For horizontal aerial and aquatic flights (trajectory <10 deg from horizontal), we assumed that the x- and y-axes were aligned with the bird's direction and, therefore, did not transform the digitized points.

To convert the linear variables to a consistent set of units, we used the body length of the bird in each frame, as determined by the distance between the eye and the distal tip of the tail in each frame. This method of calibration accounts for variability in the distance between the camera and the bird as well as any distortion of the image created as the light passed from the water to the camera. We chose to use the entire length of the body for calibration, rather than some smaller anatomical length (e.g. culmen), as both the eye and tail were highly conspicuous in all frames of the recorded videos. Visual inspection of the aquatic data revealed pronounced head movement (relative to the body) in sync with the wingbeat cycle (i.e. body length varied with position in the stroke cycle) (*Lapsansky and Tobalske, 2019*). Because of this, we smoothed the raw body-length data using the 'smoothingspline' method of fitting in MATLAB and a smoothing parameter of 1e-04 to account for the head movement of the bird. To account for digitization error of the anatomical landmarks themselves, we smoothed the kinematic data using the same 'smoothingspline' method of fitting in MATLAB using a smoothing parameter of 0.01, based on *Clifton and Biewener, 2018*.

For aerial flights, we computed wing excursion based on the elevation of the wingtip, relative to the average elevation of the eye and tail, and the wingbeat frequency as the number of complete wingbeats divided by the total duration of those wingbeats for each flight. Bird-centered stroke-plane angle was calculated as the angle between the vector describing the path of the wingtip between its minimum and maximum elevation relative to the direction the bird was traveling (*Figure 1*). We were unable to measure airspeed of alcids in aerial flight without disturbing their motion. Luckily, however, flight speeds of three of these species and their relatives have previously recorded in the wild (*Spear and Ainley, 1997*). *Spear and Ainley, 1997* categorized alcids as medium (tufted puffins, pigeon guillemots, and rhinoceros auklets) and large (common murres) (*Spear and Ainley, 1997*). To capture the full range of airspeeds exhibited by each species, we assumed alcids to have flown at the mean airspeed observed for birds of that size class flying in a crosswind ±1.96 * standard deviation of that measure (i.e. 95% prediction interval). Thus, we assumed medium alcids in our study (tufted puffins, pigeon guillemots, and horned puffins) to have flown at airspeeds between 13.95 m s$^{-1}$ and 18.65 m s$^{-1}$ and large alcids (common murres) to have flown at airspeeds between 13.32 m s$^{-1}$ and 24.68 m s$^{-1}$. We also computed $St$ based on ground speed by comparing the movement of flying alcids to stationary objects (e.g. rocks, floating debris, standing waves) in each video. We did not measure chord angle for aerial flights given the low frame rates of our aerial videos for three species. Each perpendicular aerial flight (totaling n = 18) is represented by between 4 and 46 complete wingbeats (median: 15).

For aquatic flights, wing excursion was calculated as the difference between the maximum and minimum elevation of the wrist for a given wingbeat, relative to the average elevation of the eye and tail. If anything, this is a slight underestimate of wing excursion, as the hand-wing sometimes appeared to exhibit slightly greater excursions than the wrist (≤10%). However, we chose to digitize the wrist as it was consistently visible in all videos. Frequency was the inverse of the duration of each wingbeat. Chord angle was the angle at mid-stroke between the position vector running from the wingtip to the wrist and that running from the tail to the eye. Stroke-plane angle was calculated as the angle between the bird-centered position vector describing the path of the wrist between its minimum and maximum elevation relative to the direction the bird was traveling. For aquatic flights, used the position of the tail to calculate velocity, as our previous work has demonstrated that the

head is an unreliable indicator of overall body motion in swimming alcids (*Lapsansky and Tobalske, 2019*). Details of the velocity calculation, including how we corrected for the effects of pitching in our calculation, are described in more detail in *Lapsansky and Tobalske, 2019*. The velocity due to pitching of the body was typically <5% of the translational velocity. Each perpendicular aquatic flight (n = 35; 24 horizontal and 11 descending) is represented by the values for between 1 and 6 complete wingbeats (median: 3).

To convert the final wing excursion and velocity data from body lengths to meters (for ease of comparison), we measured the length of the culmen relative to the length of the body (eye-to-tail) of 15 individuals of each species engaged in aerial flight in high-resolution images gathered from the Macaulay Library at the Cornell Lab of Ornithology. We used these data to convert from body lengths to meters for individuals in our study. The average culmen length used in this analysis (averaged from values in the Birds of North America online *Rodewald, 2015*), calculated species-specific body length, and the asset numbers for the photographs are included in the supplement (*Supplementary file 1*). Given that *St* is dimensionless, our method of converting to metric units only affects our calculations of *St* based on the airspeed reported in *Spear and Ainley, 1997*.

In addition to comparing *St* of alcids to the theoretical efficient range of 0.2 < *St* < 0.4, we also compare these data to the range for birds in cruising flight (0.12 < *St* < 0.47) reported in *Taylor et al., 2003*.

## Flights parallel to the camera view

Stroke velocity (deg s$^{-1}$) was calculated from flights made parallel to the camera view. Thus, flights were selected for analysis when birds appeared to fly horizontally and straight at or straight away from the camera (±10 deg). For all flights (n = 80), we digitized the wrist and the shoulder of each bird at the maximum and minimum elevation of each wingbeat to calculate wingbeat amplitude (deg). Stroke velocity was computed as the change in angle (deg) over the duration (sec) of the stroke. For aquatic flights, this computation was performed on a stroke-by-stroke basis. For aerial flights of common murres, horned puffins, and tufted puffins, the relatively slow frame rate meant that computing the duration of each individual stroke would provide only a coarse measurement of stroke duration. Thus, we opted to compute stroke duration for flights of these species as 0.5 * the inverse of the wingbeat frequency of that flight. We validated this approximation by computing stroke duration via both methods for the aerial flight of pigeon guillemots, finding no significant differences between the two calculations (Upstroke: *t-Stat* = 1.67, *Cohen's d* = 0.037, p=0.10, n = 70 half-strokes; Downstroke: *t-Stat* = 0.71, p=0.48, *Cohen's d* = 0.019, n = 70 half-strokes; paired t-tests). While the frame rate was relatively low for aerial flights of common murres, horned puffins, and tufted puffins (29.97 or 59.94 fps), the long exposure of the video (auto-shutter) made it relatively easy to locate the top and bottom of each stroke, as the wing briefly pauses before the turnaround. Each parallel aquatic flight is represented by between 2 and 11 complete wingbeats (median: 4) and each parallel aerial flight by between 3 and 18 complete wingbeats (median: 12).

## Data visualization and statistical analyses

We plotted data using the Gramm Toolbox from *Morel, 2018* in MATLAB and edited plots for visibility in Adobe Illustrator version 24.1.3 (Adobe Inc, San Jose, California, USA). Statistical analyses were performed using R version 3.6.3 (R Foundation for Statistical Computing, Vienna, Austria). To investigate the effect of each fluid (i.e. air or water) and type of aquatic flight (i.e. horizontal or descending) on *St* and kinematic parameters we built linear models (function *lm* in package 'stats') (e.g. *ln(KinematicVariable)~Species * Fluid*) and assessed the significance of the fixed effects using a type I ANOVA (function *anova* in package 'stats'). If the interaction between species and fluid was found to be insignificant, it was removed from the model and the model was fit again with only the main effects. To ensure normality and homoscedasticity of the residuals for each model, we log-transformed numerical data and systematically checked the diagnostic plots. We tested for the presence of outliers after each model fit using the function 'outlierTest' from the R package *car* and excluded significant outliers from analyses (*Fox and Weisberg, 2019*). We report eta-squared ($\eta^2$) calculated by the function *eta_sq* from the R package 'sjstats' (*Lüdecke, 2020*). Pairwise *post hoc* tests (for within-species differences between air and water) were performed using the *TukeyHSD*

function in R and p-values for each within species comparison are reported in alphabetical order by species name.

The stroke velocity data displayed a significant departure from homoscedasticity due to unequal variances among species. Thus, we tested for differences in stroke velocities (both upstroke and downstroke) within each species using the R function *t.test* (with *var.equal = FALSE*) and a Bonferroni-corrected critical p-value of 0.0125 (p=0.05/4 species) to account for multiple testing.

For all statistical analyses, we treated flights as independent and used the average value of the kinematic parameter exhibited for that flight for testing (rather than analyzing each wingbeat as independent). For *St*, we propagated the standard deviation in wing excursion through to calculate the standard deviation in *St*. We report means ± s.d. unless otherwise specified.

## Acknowledgements

This work was made possible by the Alaska SeaLife Center and its excellent support staff. We are grateful to Art Woods, Mark Mainwaring, Hila Chase, and Erin Keller for providing comments on an early version of this manuscript. We thank Romain Boisseau for advice on the statistical analyses. Finally, we thank Christian Rutz and Richard Bomphrey for their comments, which greatly improved the quality of this manuscript.

## Additional information

### Competing interests

Daniel Zatz: is affiliated with ZatzWorks Inc. The author has no financial interests to declare. The other authors declare that no competing interests exist.

### Funding

| Funder | Grant reference number | Author |
| --- | --- | --- |
| National Science Foundation | EFRI 1935216 | Bret W Tobalske |
| National Science Foundation | CMMI 1234737 | Bret W Tobalske |
| Drollinger-Dial Family Charitable Foundation | | Anthony Lapsansky |

The funders had no role in study design, data collection and interpretation, or the decision to submit the work for publication.

### Author contributions

Anthony B Lapsansky, Conceptualization, Data curation, Formal analysis, Investigation, Visualization, Methodology, Writing - original draft, Project administration; Daniel Zatz, Resources, Data curation, Investigation, Writing - review and editing; Bret W Tobalske, Conceptualization, Resources, Supervision, Methodology, Writing - review and editing

### Author ORCIDs

Anthony B Lapsansky https://orcid.org/0000-0001-7530-7830

### Ethics

Animal experimentation: All work was approved by the University of Montana's Institutional Animal Care and Use Committee (AUP 004-19BTDBS-020419). Work at the Alaska SeaLife Center was performed with approval from the animal husbandry and research staff.

### Decision letter and Author response

Decision letter https://doi.org/10.7554/eLife.55774.sa1
Author response https://doi.org/10.7554/eLife.55774.sa2

# Additional files

## Supplementary files

• Supplementary file 1. Average culmen length, asset numbers, and calculated body length during flight for each of four species of alcid. Calculated body lengths were used to convert from units of species-specific body length to metric units. Average culmen length was calculated as the mean of all values present in the *Birds of North America* entry (*Rodewald, 2015*) for adult birds (males and females) of that species. Multiple birds were digitized in some photographs. See Materials and methods for details.

• Transparent reporting form

## Data availability

All data are available at the following link: https://github.com/alapsansky/Lapsansky_Zatz_Tobalske_eLife_2020 (copy archived at https://github.com/elifesciences-publications/Lapsansky_Zatz_Tobalske_eLife_2020).

The following datasets were generated:

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
