## [Decision Letter]

**Acceptance summary:**

The challenging measurements the authors have taken make a valuable contribution to the field. Their analyses raise many interesting questions that can be tested in the near future, regarding the kinematic and anatomical changes that are required for the transition from air to water and back again at the species level (evolutionary change) and at the individual level (instantaneous behavioural change).

**Decision letter after peer review:**

Thank you for submitting your article "Alcids 'fly' at efficient Strouhal numbers in both air and water but vary stroke velocity and angle" for consideration by *eLife*. Your article has been reviewed by two peer reviewers, one of whom is a member of our Board of Reviewing Editors, and the evaluation has been overseen by Christian Rutz as the Senior Editor. The reviewers have opted to remain anonymous.

The reviewers have discussed their reviews with one another and the Guest Editor has drafted this decision letter, to help you prepare a revised submission.

We would like to draw your attention to changes in our revision policy that we have made in response to COVID-19 (https://elifesciences.org/articles/57162). Specifically, we are asking editors to accept without delay manuscripts that they judge can stand as *eLife* papers without additional data, even if they feel that they would make the manuscript stronger. Thus the revisions requested below only address clarity and presentation. However, the points have been deemed essential prior to acceptance.

Summary:

The reviewers agreed that this manuscript presents interesting new data and conclusions on a topic that is of active interest; it makes a useful contribution. The manuscript comprises impressive scope and a large amount of careful work. The data are collected from free-living animals, which is difficult, and very important for our understanding of real animal locomotion. The laboratory-like facility in Alaska for underwater filming of captive birds is a wonderful resource and its use is allied with filming of free flying pigeon guillemots from land in Monterey and following murres and puffins with a helicopter. Data are presented in clear, convincing and attractive figures. The manuscript raises interesting hypotheses for further work, including regional specialisation of muscles inserting on the sternum that are recruited differentially when the stroke angle changes between media. Currently, however, the paper requires some work before being of publishable standard.

Essential revisions:

1) We request that the statistics and reporting of the statistical model outputs are strengthened. The statistical analyses are under-reported and the approach not justified satisfactorily.

– All statistical information should be given for all tests. The value of the test statistic (t , F etc.) and an effect size (for example, r2 or partial eta) must be stated, as well as the corresponding p-value.

– The statistical output for the species (random effect) needs to be given and a robust justification for including species as a random effect. Species is surely a fixed effect.

– When testing for differences between angular velocity in downstroke and upstroke (as illustrated in Figure 3), and wingbeat amplitude, and wingbeat frequency (Figure 4), it would be more proper to conduct a single test than multiple t-tests (i.e., Kinematics variable = fluid medium * species). If the interaction term is not significant, it can be removed from the model and the model run again with the main effects only. A similar approach to the chord angle with depth data would allow the authors to conclude for certain that the downstroke chord angle increase with descent angle is due to the tufted puffin data only. Post hoc tests may be used to identify which groups are responsible for any differences detected.

– Multiple statistical tests are conducted. There needs to be a justification of why the p-value is not adjusted accordingly (for example, HSD or Bonferroni correction).

2) Please confirm that the birds are in a cruising flight/swim. The *St* concept holds only in cruising flight/swimming, so a statistical comparison of the velocities at the beginning and end of each analysed clip is required to ensure that this is the case. This should be done for flight swimming data. In the absence of actual velocities pixels/s can be used as the metric.

3) Please reduce the amount of speculative and discursive text. The paper is very strong without some of the sections that overreach. This means:

– reducing parts of the Introduction dealing with the jack of all trades contradictions (although we suspect some will need to remain to show the existing dogma, providing context for the new work);

– stating the interesting hypotheses raised for future work succinctly, since the current work does not provide any direct support with which to test them (muscle fibre types; attachment points; gearing).

– Introduction: It is well established which wing shape(s) provide the most efficient flight. Birds that utilise both water and air for transport have very different wing shapes to those of aerial birds. The trade-off is already very evident. Further, high-speed aerial flight is not always the most energy efficient way to cover distance – it is more likely the best option for a bird with short wings of low area: i.e. high wing loading. A brief comment should be added on trade-offs in wing shape for air and water specialists: specifically, the consequences/result of high wing loading.

4) Add a comment on the robustness of *St* to large changes in kinematics and what it means for an animal, energetically (since metabolic cost is the price they pay), to be in the high or low region of the 'efficient zone'. Discuss whether speed is a physical consequence of f*a (rather than a third independently selected parameter) leading to an inevitable convergence on a *St* number range.

---

## [Author Response]

Essential revisions:1) We request that the statistics and reporting of the statistical model outputs are strengthened. The statistical analyses are under-reported and the approach not justified satisfactorily.– All statistical information should be given for all tests. The value of the test statistic (t , F etc.) and an effect size (for example, r2 or partial eta) must be stated, as well as the corresponding p-value.– The statistical output for the species (random effect) needs to be given and a robust justification for including species as a random effect. Species is surely a fixed effect.– When testing for differences between angular velocity in downstroke and upstroke (as illustrated in Figure 3), and wingbeat amplitude, and wingbeat frequency (Figure 4), it would be more proper to conduct a single test than multiple t-tests (i.e., Kinematics variable = fluid medium * species). If the interaction term is not significant, it can be removed from the model and the model run again with the main effects only. A similar approach to the chord angle with depth data would allow the authors to conclude for certain that the downstroke chord angle increase with descent angle is due to the tufted puffin data only. Post hoc tests may be used to identify which groups are responsible for any differences detected.– Multiple statistical tests are conducted. There needs to be a justification of why the p-value is not adjusted accordingly (for example, HSD or Bonferroni correction).

These are all fair critiques and we have done our best to correct these issues.

We report proper statistics for each model. We also removed a number of significance tests, as many were unneeded for testing our hypotheses.

When testing for significant differences in kinematic parameters across species, we specify our models as *Kinematic variable = Species * Condition*. Thus, both species and condition are treated as fixed effects now. Because we have an unbalanced design (fewer descending flights than horizontal, for example), the order of the variables unfortunately matters. Based on the book “Practical Regression and ANOVA using R” by Julian Faraway, specifying our models with *Species* first allows us to determine if the relationship between *Kinematic variable* and *Condition* is significant, after accounting for the variation due to *Species*. In other words, the ANOVA function compares the model *Kinematic variable = Species* to the model *Kinematic variable = Species + Condition,* and then *Kinematic variable = Species + Condition* to *Kinematic variable = Species + Condition + Species:Condition.*

If you disagree with this method of testing, we are happy to change things!

Within species tests are now performed using either the TukeyHSD() function or a Bonferroni-corrected p-value. Details are included in the text.

2) Please confirm that the birds are in a cruising flight/swim. The St concept holds only in cruising flight/swimming, so a statistical comparison of the velocities at the beginning and end of each analysed clip is required to ensure that this is the case. This should be done for flight swimming data. In the absence of actual velocities pixels/s can be used as the metric.

This is a great point! The translational velocity in swimming is naturally pulsatile, so the speeds will not be perfectly consistent, but you are correct that they should not exhibit any trends among wingbeats. Before including flights in the data set, we plotted the velocity (in body lengths per second) to verify that the animal was not consistently accelerating or decelerating. We forgot to mention this in the original text. It has been added now.

When looking at the per-wingbeat data contained in the file ‘AquaticKinData_perp.xlsx’ on the GitHub page, you can see that the velocity (in body lengths per second) does not vary more than ~1 bl s^-1^ between the wingbeat at the start of the run and the wingbeat at the end.

For aerial flights, the ground velocity was determined based on an object or standing wave under the bird, which quickly disappears from view as the camera follows the animal. We mainly discuss Strouhal number in air relative to average air speed of the species, given that we could not measure the air speed of the bird, and point out the issues with our ground speed measures. We could try to measure ground speed at the beginning and end of each flight, but it would not tell us about the consistency of the animal’s air speed. However, because the birds were flying over open water, we are reasonably justified in assuming that they were cruising.

3) Please reduce the amount of speculative and discursive text. The paper is very strong without some of the sections that overreach. This means:– reducing parts of the Introduction dealing with the jack of all trades contradictions (although we suspect some will need to remain to show the existing dogma, providing context for the new work);– stating the interesting hypotheses raised for future work succinctly, since the current work does not provide any direct support with which to test them (muscle fibre types; attachment points; gearing).– Introduction: It is well established which wing shape(s) provide the most efficient flight. Birds that utilise both water and air for transport have very different wing shapes to those of aerial birds. The trade-off is already very evident. Further, high-speed aerial flight is not always the most energy efficient way to cover distance – it is more likely the best option for a bird with short wings of low area: i.e. high wing loading. A brief comment should be added on trade-offs in wing shape for air and water specialists: specifically, the consequences/result of high wing loading.

The text has been trimmed considerably, including over 1000 words trimmed from the original Discussion.

With specific reference to the original submission, we agree that our statement was vague and, therefore, sounded suspicious. We were attempting to point out that a trade-off in flight performance caused by using the wings for locomotion in two media is not well-established, but our phrasing made it seem that we were arguing against an aerial-terrestrial trade-off altogether. Hopefully, the revised text is clearer.

On wing shapes of wing-propelled diving birds, more research is needed to adequately test whether they have different wing shapes compared with foot-propelled diving birds. Based on wing measurements from the sources mentioned in the introduction, the wing shapes (at least in the parameters typically measured) of alcids and Anatidae, including many foot-propelled diving ducks, ‘non-diving’ ducks, and geese, overlap entirely. Shearwaters, which routinely dive down to 60 m (Croll and Tershy, 2012) [deeper than most alcids], have very different wing shapes from most diving birds, as do boobies and gannets. These data have not been rigorously analyzed, so most of discussion related to the wing shapes of diving birds has been removed from the paper. But until this question has been addressed with multivariate data, in a phylogenetic context, we feel caution precludes describing the wings of wing-propelled divers as being different in shape from those of other flying birds.

4) Add a comment on the robustness of St to large changes in kinematics and what it means for an animal, energetically (since metabolic cost is the price they pay), to be in the high or low region of the 'efficient zone'. Discuss whether speed is a physical consequence of f*a (rather than a third independently selected parameter) leading to an inevitable convergence on a St number range.

Good point! We added a paragraph in the subsection “Flights perpendicular to the camera view” to discuss these topics.